# Research on the Application of NbS in Watershed Ecological Restoration: A Case Study of Jiulong River Watershed Shan-Shui Initiative

**Wei Li** [1,2] **, Rui Sun** [1,2] **and Ye Tian** [1,*]

1. Institute of Urban Environment, Chinese Academy of Sciences, Xiamen 361021, China; wli@iue.ac.cn (W.L.); rsun@iue.ac.cn (R.S.)
2. University of Chinese Academy of Sciences, Beijing 100049, China
* Correspondence: ytian@iue.ac.cn; Tel.: +86-138-1138-3881

**Abstract:** Nature-based solutions (NbS) rapidly develop globally to address societal challenges and provide human well-being and biodiversity. Watershed restoration plays an essential role in enhancing the ecological and socio-economic benefits of the region. The design and implementation of watershed restoration projects are crucial to their effectiveness, and NbS has been applied as a concept in ecosystem-related projects. This paper proposes an evaluation method to assess the implementation of the Shan-Shui Initiative in the Jiulong River Watershed restoration projects based on the eight criteria proposed by the International Union for the Conservation of Nature (IUCN) Global Standard for Nature-Based Solutions. The aim is to standardize the implementation process of watershed restoration projects to achieve more significant benefits and practically apply the concept of NbS in watershed restoration. The implementation scheme of the Shan-Shui Initiative in the Jiulong River Watershed is designed to evaluate and improve the design and implementation scheme of ecological restoration projects in the actual basin. According to the assessment results, the degree of matching based on NbS for the implementation scheme of the Jiulong River Watershed Shan-Shui Initiative is 73%, which meets the criteria of NbS but needs to be improved in terms of monitoring and assessment, synergistic management, and benefit trade-offs.

**Keywords:** watershed ecological restoration; nature-based solutions; Shan-Shui Initiative; scenario evaluation; adjustment of measures

## 1. Introduction

Nature-based solutions (NbS) have been recognized as an important tool to jointly tackle ecological restoration with an integrated and sustainable approach [1]. Furthermore, NbS is fast becoming a leading force in driving positive change across the world. For example, NbS interventions, such as a combination of restoration of wetlands, marshes, and dune systems and the creation of protected areas, are being deployed to address coastal erosion, risk of submersion, and marine biodiversity loss in coastal areas [2–4]. In Jamaica, coastal disaster risks are addressed through ecological conservation restoration projects such as protected area management, mangrove restoration, and coral restoration [5], and M. Acreman et al., argue that the social challenges of water issues in Africa, including flood disasters and water security, can be effectively addressed through ecosystem conservation and restoration [6]. However, NbS is used as a concept, and no specific measures are designed based on the framework of NbS. This situation will not be able to distinguish the difference between ecological restoration under the NbS concept and other ecological restoration and will reduce the role of NbS in the field of ecological restoration [7–10]. Thus, the International Union for the Conservation of Nature (IUCN) published the IUCN Global Standard for Nature-Based Solutions (hereafter referred to as the Global Standard) and the IUCN Global Standard for Nature-Based Solutions User Guide (hereafter referred to

as the User Guide) to provide a clear understanding of the concept of NbS and make it work in practice [8,9]. The Global Standard establishes eight guidelines to help design and standardize specific NbS measures. Twenty-eight indicators are used to assess the criteria. The User Guide proposes a traffic-light system for assessing specific measures (strong, sufficient, weak, insufficient). Items that score "insufficient" on the criteria or "inadequate" overall do not meet the NbS criteria. This tool can standardize the design and implementation of specific NbS measures [11]. However, this criterion is rarely used in actual case evaluations currently, especially in addressing watershed restoration issues.

Early civilizations developed around seasonal river floodplains, and the natural attributes of rivers remain vital to humans today [12]. However, due to climate change and large-scale land-use changes [13], as well as changes in water use and demographic pressures [14], inland watersheds on a global scale have generally experienced vegetation degradation, soil erosion, water pollution, reduced biodiversity, and water scarcity, which have caused changes in watershed ecosystem services and an increasingly apparent conflict between ecological conservation and economic development [15,16]. Ecosystem restoration and management at the watershed scale is a profitable way to solve ecological and socio-economic problems in the watershed [17–21]. Compared to single ecosystem restoration, watershed-scale ecosystem restoration needs to consider the correlation between different ecosystems within a watershed and the interaction between socioeconomic and ecosystem impacts [22,23]. This places high demands on the design of restoration or management programs, which must be designed according to the restoration goals regarding technical methods, management tools, funding, and policy support [24]. This process requires the participation of policymakers, managers, recovery practitioners, and stakeholders [25–28]. Whether the design and implementation plan of the ecological restoration of the watershed is reasonable will directly affect the effect produced by the restoration [29].

Due to the frequent need for watershed ecological restoration projects to be planned and designed at larger scales, which aligns with the requirement for NbS measures to be implemented at the landscape scale, this paper proposes an application concept for NbS in actual watershed ecological restoration [30]. It is crucial to reach a consensus on what constitutes successful and sustainable NbS [30]. Clear and coherent principles and standardized evidence-based frameworks are needed [31]. This will enable NbS to be designed and implemented using the best evidence-based criteria The Global Standard proposes eight guidelines used to ensure that the NbS concept is clearly described, understood, implemented, and communicated so that NbS deliver their intended outcomes and also to guide and regulate the design and implementation of ecosystem-related measures. The design or implementation plan of watershed ecological restoration projects is evaluated according to the eight guidelines proposed by the Global Standard. The implementation plan is supplemented or adjusted according to the evaluation results, thus regulating the implementation process of watershed ecological restoration. The aim is to enhance the role of the project in addressing social challenges, strengthening ecosystem integrity and biodiversity, enhancing the overall stability of the project, and bringing more significant benefits.

## 2. Introduction to the Concept of Shan-Shui Initiative

The Landscape, Forest, Field, Lake, and Grass Ecological Protection and Restoration Project (from now on referred to as Shan-Shui Initiative) is an ecological protection and restoration project carried out by China by the concept of "Landscape, Forest, Field, Lake and Grass is a community of life" [32]. This project has changed from the ecological project of protecting and restoring a single element in the past (e.g., the Three Northern Protection Forests Project, the Returning Farmland to Forest Project) to a large-scale ecological restoration project with multiple elements, mainly at the regional or watershed scale [25–28]. The main purpose of the project is to solve the problems of water environment destruction, mine ecosystem degradation, soil erosion, and farmland ecosystem degradation in some small watersheds through ecological restoration.

China emerges as a leading proponent of nature-based solutions (NbS) to improve global environmental management [33]. The Ministry of Natural Resources of China, together with the Ministry of Finance and the Ministry of Ecology and Environment, jointly issued the "Guidelines for Ecological Protection and Restoration Projects in Mountain, Water, Forest, Field, Lake, and Grass", which cites the concept of NbS in the terminology and definitions section [34]. Although these projects have achieved specific results, environmental protection and restoration of mountains, water, forests, fields, lakes, and grasses are being explored and perfected. Many problems have been exposed, such as insufficient synergy mechanism of engineering projects, lack of integrity between projects, imperfection of related systems, and insufficient public participation. Irregular engineering design is one of the reasons for these problems [35–37].

## 3. Research Method

The study methodology uses the NbS self-assessment tool developed in the User Guide. The tool allows scoring the degree of f conformity of the eight criteria according to the scheme of the specific measures. The degree of criterion matching score needs to be calculated based on the f-conformity score of the criterion's corresponding indicator. Ultimately, the criterion score can be used to calculate the overall degree of NbS compliance of the program. Scores for the degree of conformity were given in the form of percentages for scores of $\geq 75$, $\geq 50$ and $<75$, $\geq 25$ and $<50$, and $<25$ for Strong, Adequate, Partial, and Insufficient, respectively. If one criterion was assessed as Insufficient or the result of the overall degree of match for NbS was Insufficient, then the intervention program did not meet the concept of NbS. In terms of adjustments and enhancements to the implementation program based on the assessment results, the score for the overall degree of match for NbS is used as the basis for whether the implementation program meets the NbS concept. The degree of match for each criterion is used as a basis for whether the implementation needs to be supplemented or adjusted about the content of that criterion. The degree of match of an indicator can provide the specific elements of the guideline that need to be adjusted (Figure 1). In evaluating specific watershed ecological restoration projects through this method, our team found that the tool needed to be more challenging to use to score the degree of matching of indicators quantitatively. The User Guide only gives quantitative scoring criteria for specific indicators, resulting in highly subjective final results with a sufficient basis to confirm the credibility of the scores [8,9]. This is achieved by refining the indicators into pertinent questions based on the corresponding descriptions of each indicator in the guidelines (Figure 1). These questions should directly relate to the indicators and seek to capture their essence and significance. Ultimately, the degree of indicator alignment is assessed by calculating the percentage of questions that the watershed ecological restoration project satisfies. This provides a clear and objective measure of the project's compliance with the indicators, enabling decision makers to determine its overall performance and effectiveness. By considering the percentage of questions that are addressed and satisfied, it becomes possible to identify any potential shortcomings or challenges that need to be addressed for improved sustainability and overall success. Therefore, refining the indicators into relevant questions, assessing the degree of indicator alignment, and considering the percentage of questions satisfied is an effective approach to evaluating the performance and sustainability of watershed ecological restoration projects. This approach ensures that decision makers have a comprehensive understanding of the project's achievements, areas for improvement, and potential impact on the environment, society, and economy (Table 1). To verify the feasibility of the method, the team evaluated the implementation plan of the Jiulong River Watershed Shan-Shui Initiative through this method. To ensure that the results are credible, the participants involved in scoring included those who developed the program as well as researchers in related fields. The specific calculation method is as follows.

$$S_I = \frac{q}{z},$$

$$S_C = \frac{\sum_{i=1}^{n} I_i}{n},$$

$$S_T = \frac{\sum_{j=1}^{m} C_j}{m},$$

where $S_I$ is the index matching score; $S_C$ is the criterion matching score; $S_T$ is the overall NbS matching score; $I_i$ is the i-th index score; $C_j$ is the *j*-th criterion score; q is the number of individual index matching questions; z is the total number of individual index questions; n is the number of indexes; m is the number of criteria.

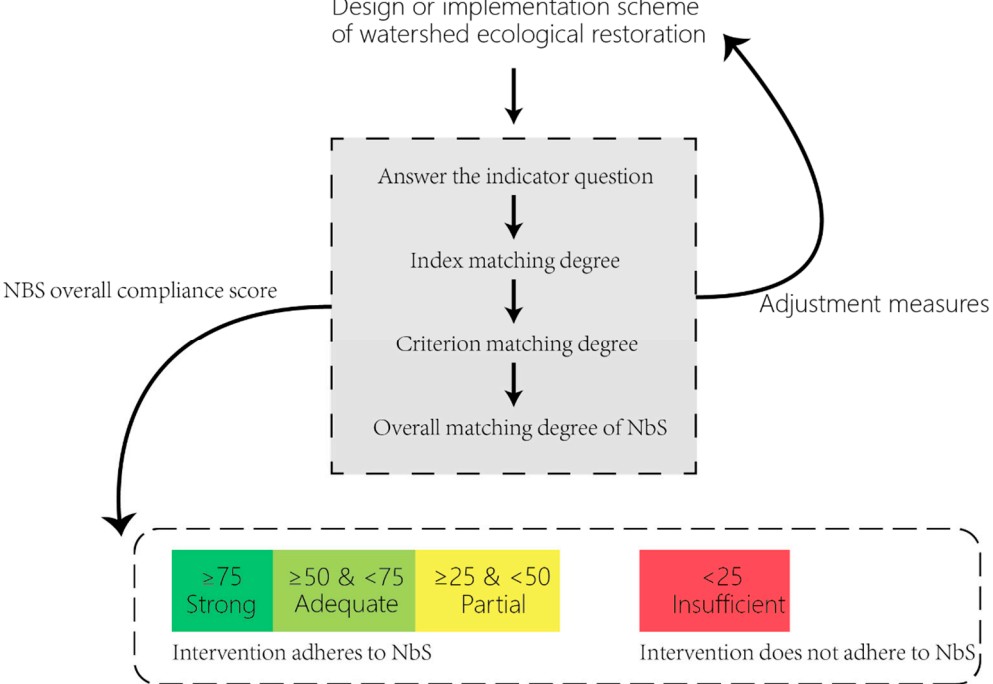

**Figure 1.** Evaluation process.

**Table 1.** The method of quantifying indicators is exemplified by indicator c1.1.

| C-1.1: The Most Pressing Societal Challenge(s) for Rights-Holders and Beneficiaries are Prioritized | Matching Degree: 67% |
|---|---|
| **Index question:** | |
| Whether interventions are aimed at addressing more than two social challenges defined by NbS, including climate change mitigation and adaptation, disaster prevention and mitigation, economic and social development, human health, food security, water security, ecological degradation, and loss of biodiversity | YES |
| (1) Whether the social challenges of power holders and beneficiaries in the region have been identified | NO |
| (2) Whether to weigh and compare the impact of social challenges on society and give priority to dealing with the most urgent social challenges | NO |
| **Question feedback:** | |
| (3) Part of identification and diagnosis of the problem: the identified problems correspond to four types of social challenges: disaster prevention and mitigation, economic and social development, water security, disaster prevention and mitigation, eco-environmental degradation, and loss of biodiversity | |
| (4) Part of identification and diagnosis of the problem: the social challenges identified include those that power holders and beneficiaries need to address, such as water security, disaster prevention, and mitigation. | |
| (5) The feasibility analysis of the importance of the implementation plan: There is no analysis and tradeoff between the identified social challenges and the urgency of ecosystem problems | |

## 4. Case Overview

The Jiulong River Basin is an important governance area in the Shan-Shui Initiative. The Jiulong River Watershed is located in the southeastern coastal area of Fujian Province, China, with coordinates from 116°47′ to 118°02′ and latitude from 24°13′ to 25°51′ N. The

basin covers an area of 14,837 square kilometers, accounting for about 12% of the land area of Fujian Province. The basin involves 5 cities and 19 counties. The coastal zone is the most economically developed, most open to the outside world, and most densely populated region in China. The Jiulong River Basin plays a very important role in maintaining the stability of the regional ecosystem, water conservation, and biodiversity. It is an important component of the ecological security barrier along the southeast coast of China and is also a key area for biodiversity conservation in China. It has an important position in the national ecological security strategy (Figure 2).

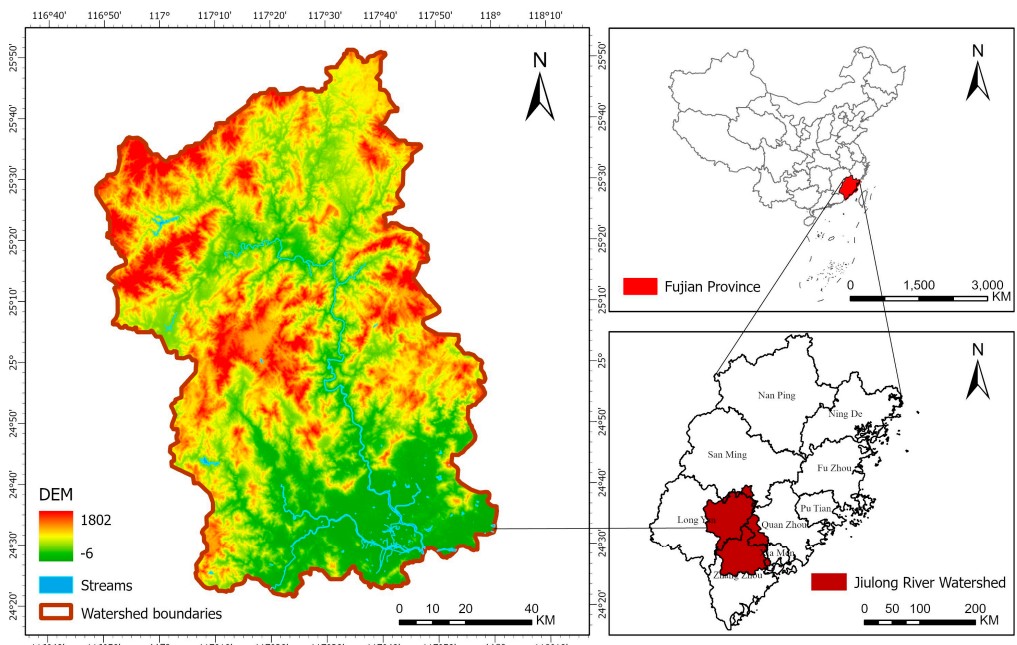

**Figure 2.** Location and map of Jiulong River Basin.

The Jiulong River Watershed Shan-Shui Initiative was successfully declared in 2021, with a total project investment of 7.861 billion yuan (including 2 billion yuan of central financial award funds) and an implementation period of 2021–2023, which is currently in the process of implementation. The overall goal is to achieve the following by 2023: the water quality of the main watersheds in the project area reaches or exceeds the national standard. The proportion of water and soil loss is better than Class III, which is equal to or greater than 95.4% (including the proportion of Class I and Class II water, which is equal to or greater than 60%). The rate is less than or equal to 8%, the forest stock volume is more than or equal to 70 million cubic meters, and the biodiversity is significantly improved and so on. The implementation period of the project is from 2021 to 2023, which will be implemented in three years. The integrated protection and restoration project of the landscape, forest, farmland, lake, grassland, and sand in the Jiulong River Basin aims to restore forest ecological areas of 8300 acres in 2021, 100,000 acres in 2022, and 8000 acres in 2023. The comprehensive river management project aims to restore a length of 60 km in 2021, 120 km in 2022, and 120 km in 2023.

This article's case-related information is mainly for the "Fujian Jiulong River Watershed Mountain, Water, Forest, Lake, Grass, and Sand Integration Protection and Restoration Project Implementation Plan" issued by the Department of Ecology and Environment of Fujian Province, China. The program consists of the following five main components:

1. The section on the basic situation of the basin includes the investigation of the ecological and socio-economic status of the region, the identification and diagnosis of regional ecological problems, and the analysis of the importance and feasibility of the project.

2.  In the project implementation part, the main contents include the overall objectives and effectiveness assessment indicators of the project implementation and subproject objectives and performance indicators, the layout and implementation period of the project, the implementation content, and the technical route.
3.  Project estimates and funding channels section.
4.  Organization and implementation and supervision and management part, mainly including the security measures of the project and management methods.
5.  The benefit analysis section, which analyzes the expected benefits generated by the three ecological–social–economic aspects of the project.

## 5. Evaluation Results and Analysis

The following results were obtained from evaluating the implementation plan of the Jiulong River Watershed Shan-Shui Initiative through the methodology proposed in this paper. A visual pie chart with the size of the area of the different colored blocks representing the level of match for each criterion shows that the scheme has an overall match score of 73% for NbS, and no criterion assessment results in Insufficient, indicating that the project is mainly compatible with the NbS concept. In terms of criteria, three criteria were assessed as Strong, respectively, criterion 3, criterion 4, and criterion 8. Criterion 3 had the highest level of match at 94 percent, while the matching degree of criterion 5 was the lowest, at 43%; four criteria were assessed as Adequate, respectively, criterion 1, criterion 2, criterion 6, and criterion 7, and only criterion 5 was assessed as Partial. As for the indicators, 16 out of 28 were assessed as Strong, and all indicators of criterion 3 were high matches: 4 as Adequate; 5 as Partial; and 3 as Insufficient. (Figure 3). The results show that the Shan-Shui Initiative is an NbS measure that, if successfully implemented, can positively address social challenges, enhance ecosystem integrity and biodiversity, and achieve sustainability in the basin. However, at the same time, the program still has room for improvement in some criteria. The following is a specific analysis of the evaluation results (Table 2; Figure 3).

Criterion 1 requires interventions to identify and address societal challenges (Table 2). NbS defines seven societal challenges that need to be addressed by interventions, namely climate change mitigation and adaptation, disaster risk reduction, ecological degradation and biodiversity loss, human health, socioeconomic development, food security, and water security. The guideline match score was 67% (Table 1), and the three indicators associated with it were assessed as Adequate, Strong, and Partial. The effects of the Jiulong River Watershed Shan-Shui Initiative regarding the listed IUCN societal challenges can be positive to some extent. For instance, it could lead the development of regional green eco-industries that can best fulfill its promise in terms of contribution to economic growth [38]. However, NbS interventions are multifaceted and highlight the fundamental influences that preservation and diversification of ecosystems can have on human well-being [39]. Indicator c1.3 (Table 2), which requires interventions to establish human well-being-related targets and to evaluate them regularly, could have been better matched. The results indicate that the implementation program needs to be improved in the establishment and assessment of human well-being-related goals, mainly because there are no human well-being-related performance indicators in the main goal section of the implementation program.

Criterion 2 requires interventions to be designed at the scale of the landscape, and the score for this criterion's degree of match is 69% (Table 2). Three indicators related to this criterion are Strong, Strong, and Insufficient. Indicator c2.3 reduces the matching level of this criterion, mainly because of the need for a monitoring and assessment program and risk management measures for the area around the project implementation in the implementation section. Although an environmental monitoring and management system has been established for the Jiulong River Watershed Shan-Shui Initiative, the system mainly monitors and manages the environment within the project implementation area and lacks environmental assessment of the site's surrounding area. Based on the NbS concept, the impact generated by ecosystem change often radiates to the surrounding area, so if the

impact of ecological restoration on the surrounding area needs to be taken into account, it may increase unexpected risks. For example, a case study from a drought-prone region in rural India shows that empowering young people through agroforestry provides them with the opportunity to be gainfully employed in their field, rather than migrating to cities or urban centers [40].

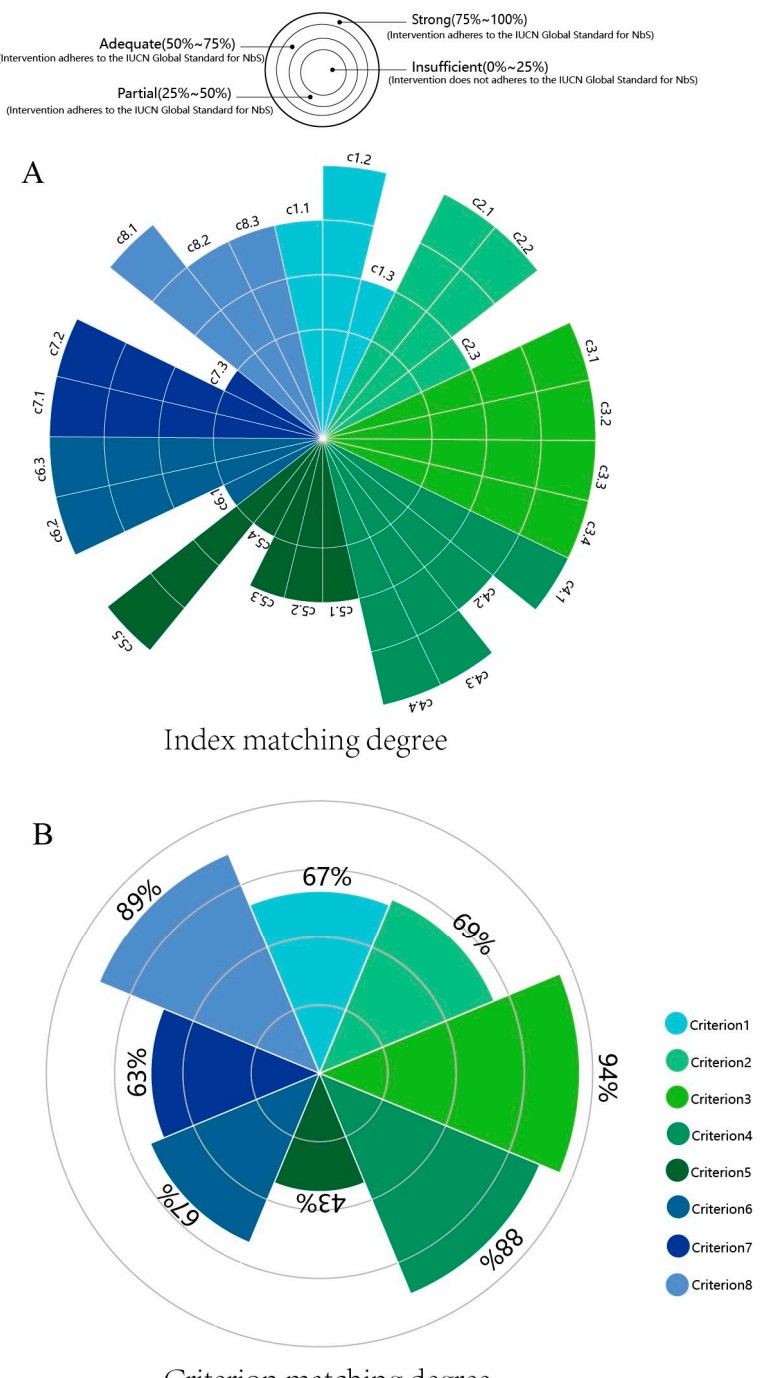

**Figure 3.** Assessment results of the Jiu Long Jiang Shan-Shui project based on NbS indicators and the degree of guideline matching. (**A**) degree of matching of indicators; (**B**) degree of matching of criterion. IUCN: International Union for the Conservation of Nature; NbS: Nature-based solutions.

**Table 2.** The Jiulong River Watershed Shan-Shui Initiative's score on the NbS overall indicators and criterion match.

| Overall Match of NbS | Adequate 73% |
| --- | --- |
| Criterion 1: NbS effectively address societal challenges | Adequate (67%) |
| C-1.1 The most pressing societal challenge(s) for rights-holders and beneficiaries are prioritized | Adequate (67%) |
| C-1.2 The societal challenge(s) addressed are clearly understood and documented | Strong (100%) |
| C-1.3 Human well-being outcomes arising from the NbS are identified, benchmarked, and periodically assessed | Partial (33%) |
| Criterion 2: Design of NbS is informed by scale | Adequate (69%) |
| C-2.1 The design of the NbS recognizes and responds to interactions between the economy, society, and ecosystems | Strong (75%) |
| C-2.2 The design of the NbS is integrated with other complementary interventions and seeks synergies across sectors | Strong (100%) |
| C-2.3 The design of the NbS incorporates risk identification and risk management beyond the intervention site | Partial (33%) |
| Criterion 3: NbS result in a net gain to biodiversity and ecosystem integrity | Strong (94%) |
| C-3.1 The NbS actions directly respond to evidence-based assessment of the current state of the ecosystem and prevailing drivers of degradation and loss | Strong (86%) |
| C-3.2 Clear and measurable biodiversity conservation outcomes are identified, benchmarked, and periodically assessed | Strong (88%) |
| C-3.3 Monitoring includes periodic assessments of unintended adverse consequences on nature arising from the NbS | Strong (100%) |
| C-3.4 Opportunities to enhance ecosystem integrity and connectivity are identified and incorporated into the NbS strategy | Strong (100%) |
| Criterion 4: NbS are economically viable | Strong (88%) |
| C-4.1 The direct and indirect benefits and costs associated with the NbS, who pays and who benefits, are identified and documented | Strong (83%) |
| C-4.2 A cost-effectiveness study is provided to support the choice of NbS including the likely impact of any relevant regulations and subsidies | Adequate (67%) |
| C-4.3 The effectiveness of the NbS design is justified against available alternative solutions, taking into account any associated externalities | Strong (100%) |
| C-4.4 NbS design considers a portfolio of resourcing options such as market-based, public sector, voluntary commitments, and actions to support regulatory compliance | Strong (100%) |
| Criterion 5: NbS are based on inclusive, transparent, and empowering governance processes | Partial (43%) |
| C-5.1 A defined and fully agreed upon feedback and grievance resolution mechanism is available to all stakeholders before an NbS intervention is initiated | Partial (33%) |
| C-5.2 Participation is based on mutual respect and equality, regardless of gender, age, or social status, and upholds the right of Indigenous Peoples to Free, Prior, and Informed Consent (FPIC) | Partial (33%) |
| C-5.3 Stakeholders who are directly and indirectly affected by the NbS have been identified and involved in all processes of the NbS intervention | Partial (33%) |
| C-5.4 Decision-making processes document and respond to the rights and interests of all participating and affected stakeholders | Insufficient (17%) |
| C-5.5 Where the scale of the NbS extends beyond jurisdictional boundaries, mechanisms are established to enable joint decision making of the stakeholders in the affected jurisdiction | Strong (100%) |
| Criterion 6: NbS equitably balance trade-offs between achievement of their primary goal(s) and the continued provision of multiple benefits | Adequate (67%) |
| C-6.1 The potential costs and benefits of associated trade-offs of the NbS intervention are explicitly acknowledged and inform safeguards and any appropriate corrective actions | Insufficient (0%) |
| C-6.2 The rights, usage of, and access to land and resources, along with the responsibilities of different stakeholders, are acknowledged and respected | Strong (100%) |
| C-6.3 The established safeguards are periodically reviewed to ensure that mutually agreed trade-off limits are respected and do not destabilize the entire NbS | Strong (100%) |
| Criterion 7: NbS are managed adaptively, based on evidence | Adequate (63%) |
| C-7.1 An NbS strategy is established and used as a basis for regular monitoring and evaluation of the intervention | Strong (100%) |
| C-7.2 A monitoring and evaluation plan is developed and implemented throughout the intervention lifecycle | Strong (88%) |
| C-7.3 A framework for iterative learning that enables adaptive management is applied throughout the intervention lifecycle | Insufficient (0%) |
| Criterion 8: NbS are sustainable and mainstreamed within an appropriate jurisdictional context | Strong (89%) |
| C-8.1 The NbS design, implementation, and lessons learnt are shared to trigger transformative change | Strong (100%) |
| C-8.2 The NbS informs and enhances facilitating policy and regulation frameworks to support its uptake and mainstreaming | Adequate (67%) |
| C-8.3 Where relevant, the NbS contributes to national and global targets for human well-being, climate change, biodiversity, and human rights, including the United Nations Declaration on the Rights of Indigenous Peoples (UNDRIP) | Strong (100%) |

Criterion 3, which requires interventions to bring about net biodiversity growth and ecosystem integrity, scored 94% compliance (Table 2), the highest of the eight criteria. The results for the four corresponding indicators were Strong, indicating that the Shan-Shui Initiative implementation program was designed to comply highly with the NbS criteria for bringing about net biodiversity growth and ecosystem integrity. A portfolio of large-scale

ecological restoration programs has been implemented in China to combat the land-system sustainability emergency, and since the 1980s, net biodiversity growth and ecosystem integrity have been the focus of China's ecological restoration programs [38]. Meanwhile, this program has established programs and measures for surveying the status of ecosystems in the watershed, setting biodiversity targets, long-term monitoring and assessment of ecosystems, and increasing ecosystem connectivity.

Criterion 4 requires interventions to be economically feasible, with a matching degree score of 88% (Table 2), and the four associated indicators are assessed as Strong, Adequate, Strong, and Strong, indicating that the implementation program is highly designed to meet the NbS criteria in terms of funding. Funding for NbS activities, such as infrastructure development, comes from public, private, bilateral, or multilateral agencies [41]. While indicator c4.2 still has room for improvement, the indicator requires measures to be a cost-effective approach while considering the impact of relevant regulations and subsidies on the cost of the measure. Although the implementation plan adopts a cost-effectiveness approach and budgets the project investment while estimating the ecological, social, and economic benefits, it needs to include measuring long-term costs, such as the cost of post-management and maintenance, which is important for continued development. In QuyNhon, economically viable and organizational contexts make them successfully carried-out major streambank restoration works on their local river [42].

Criterion 5, which requires measures based on inclusive, transparent, and empowering governance processes, scored a 43% match (Table 2). Five indicators are associated with it, being Partial, Partial, Partial, Insufficient, and Strong. This guideline has the lowest score among the eight guidelines, indicating that the design of the implementation plan for the governance process could be better, mainly reflected in establishing relevant joint mechanisms and establishing dispute resolution and stakeholder participation. The main reasons for this lack of design are specific reasons.

1. No corresponding complaint, feedback, or dispute mechanism is established in the organization, implementation, supervision, and management part. Although stakeholders' interests are protected by law, establishing relevant feedback and complaint mechanisms can significantly save time and cost so that groups whose interests have been lost can resolve disputes as soon as possible while preventing the expansion of the losses suffered.
2. The implementation plan needs to reflect the consultation process with stakeholders, especially residents, prior to the implementation of the project.
3. It only ensures stakeholders' participation in part of the project process. The residents are the primary beneficiaries of the project. Although the residents' satisfaction survey will be used as the performance assessment index after the project is completed, there are no measures to involve the stakeholders in the planning, design, and implementation stages of the project, which will result in the opinions of the beneficiaries not being adopted in the first place.

Criterion 6, which requires interventions to make fair trade-offs between primary objectives and multiple other benefits, has a matching score of 67% (Table 2), against which three indicators are assessed as Insufficient, Strong, and Strong. Based on the results, implementation programs need to be adjusted or supplemented based on indicator C6.1, which requires a cost–benefit-based assessment of different programs based on cost–benefit trade-offs. The main reason for the mismatch in indicator C6.1 is that the restoration model selection section of the implementation plan needs to weigh the benefits of different restoration models. Nbs needs measures to weigh the benefits of different approaches, including addressing social challenges and enhancing biodiversity, as well as the additional economic and social benefits that different approaches may bring.

Criterion 7 requires interventions to be evidence-based for adaptive management, with a compliance score of 63% (Table 2). The guideline is associated with three indicators: Strong, Strong, and Insufficient. The mismatch is C7.3. The mismatch is because the implementation program does not have a relevant iterative learning framework. NbS

adaptively needs interventions for adaptive management to establish an iterative learning framework to learn and accumulate experience and lessons learned in the process of adaptive management's continuous adjustment of measures.

Criterion 8, which requires that interventions be sustainable and mainstreamed in the jurisdiction, was met with a score of 89% (Table 2). It is important to note that this guideline assesses the Shan-Shui Initiative rather than specific implementation programs, as it focuses on assessing the impact of interventions at the national and global levels. The Shan-Shui Initiative is a China-wide ecological conservation and restoration initiative that is currently the mainstay of China's ecological restoration program. It has positive implications for sustainable development and social challenges. At the global level, the Shan-Shui Initiative has been selected as one of the first ten "World Ecological Restoration Flagship Projects" by the United Nations, providing experience in solving common ecological and environmental problems worldwide.

## 6. Proposed Adjustments to the Implementation Plan for the Jiulong River Watershed Shan-Shui Initiative

Through the results and analysis of the guideline-based assessment of the implementation plan of the Jiulong River Watershed Shan-Shui Initiative, the plan can be optimized in the following three main aspects.

In terms of monitoring and assessment, human well-being-related indicators should first be developed. Common frameworks for assessing human well-being include the Physical Quality of Life Index (PQoL) and the Human Development Index (HDI) [43,44]. The Millennium Ecosystem Assessment report presents a framework for assessing human well-being, which describes human well-being in five dimensions: "security, basic material needs, health, good social relations, and freedom of choice and action" [45]. These can be used as quantitative human well-being-related indicators to be included in monitoring and evaluation programs. Secondly, a monitoring and assessment program for the risks around the project should be developed. NbS requires the project to avoid negative impacts on stakeholders and ecosystems outside the site. The monitoring program should include monitoring essential ecosystems around the implemented works and the risk assessment of the surrounding area. Finally, there is a need to strengthen the assessment of economic and social aspects. The only relevant indicators of socio-economic aspects in the Jiulong River Watershed Shan-Shui Initiative program are eco-industrial development, public awareness of environmental protection, and public satisfaction. It is difficult to measure the socio-economic impact of the project by a single indicator. The program can be supplemented with socio-economic-related indicators such as employment rate, per capita income, and social capital participation.

Regarding collaborative governance, the implementation plan needs to be improved by establishing relevant participation mechanisms and safeguards and supporting and encouraging stakeholders to participate in the project. The implementation plan needs to reflect the measures for stakeholder participation in the design and implementation of the project. Only a survey on public satisfaction will be conducted after the project is completed, which is insufficient to incorporate practical stakeholder suggestions, especially residents' opinions, during the project implementation process. The study shows more benefits can be obtained by establishing a cooperation mechanism with stakeholders in watershed ecosystem management [46]. Although the different levels of management departments involved in the Jiulong River Watershed Shan-Shui Initiative have set up a working group, which has established a foundation for cross-regional cooperation, the working group is mainly composed of project managers without the participation of more stakeholders. Inviting representatives of different stakeholders to participate in the whole process of the project will help protect the stakeholders' interests and reduce unnecessary contradictions in the implementation of the project. The key to collaborative governance is to establish collaborative structures such as working groups or alliances, and different collaborative structures will have different impacts on measures. An effective synergy model requires

consultation mechanisms, and different stakeholders must play their respective roles to have a more beneficial impact [47]. Government departments are essential in establishing cooperation models and increasing stakeholder acceptance in this process [48]. When evaluating the effectiveness of the project, the equity and sustainability of NbS can also be enhanced by establishing a participatory evaluation framework [49].

Regarding benefit trade-offs, NbS requires interventions to trade off between multiple benefits. The trade-off between different benefits is missing in the implementation scheme of the Jiulong River Watershed Shan-Shui Initiative. In the restoration model selection part of the program, the relationship between the roles of different ecosystem elements in the watershed was analyzed mainly according to the theory of ecosystems. Restoration models such as ecosystem protection, natural restoration, assisted regeneration, and ecological reconstruction are adopted in different watershed areas. The choice of this model is feasible for ecological restoration projects aimed at restoring ecosystem functions. However, the NbS trade-off process needs to consider the reasonable distribution of ecological, social, economic, and other benefits at the exact cost and weigh in the dimensions of time-space and ecosystem reversibility (time dimension refers to the relative speed of impact occurrence, spatial dimension refers to the possibility that the effect can be realized locally or remotely, and when the disturbance event stops, a damaged ecosystem service may return to its original state). At present, there are many international studies on related benefit trade-offs. Li Meng di et al. provide a multi-objective trade-off method, incorporating ecological security assessment and multi-objective optimization (MOO) into the design of the watershed ecological restoration and using the assessment framework established by the pressure-state-function-response (PSFR) model to weigh multiple benefits such as ecology, economy, and society [50]. Bush J. et al. identified five key trade-offs for NbS in improving urban resilience: time; space; function; social equity; and species [51].

### 7. Conclusions

This paper evaluates the implementation plan of the Jiulong River Watershed Shan-Shui Initiative through the eight criteria proposed in the global guide and concludes the following. The implementation plan of the Jiulong River Watershed Shan-Shui Initiative complies with the NbS global guidelines. If successfully implemented by the implementation plan, the project can positively address regional social challenges, enhance biodiversity, and bring about sustainable development. However, there is still room for improvement in the monitoring and evaluation, collaborative governance, and benefit trade-off of the program, and the following suggestions for supplementation and improvement are put forward for the program based on the evaluation results.

1. In terms of monitoring and assessment, there is a need to include monitoring and assessment indicators related to human well-being. NbS interventions are comprehensive and emphasize the fundamental impacts that the preservation and diversification of ecosystems can have on human well-being. These range from climate regulation (e.g., [52]) and limiting the impacts of natural disasters such as flooding (e.g., [53]) and epidemic disease outbreaks to promoting improved human physical health and mental health. Regular assessment of the indicators is introduced to understand the impact of the project on the local population and to avoid negative impacts on the quality of life and well-being of the local population.

2. Strengthen the monitoring of the important ecosystems surrounding the implementation area to ensure that the ecological restoration project has a positive impact on the ecosystem of the entire watershed and avoids potential damage to the ecosystem, as well as monitoring programs and ecological risk assessment and management of essential ecosystems around the region. Supplement the social and economic benefits indicators and incorporate them into the monitoring and evaluation program.

3. In terms of collaborative governance, establishing synergistic mechanisms to ensure stakeholder participation at all stages of the project promotes shared governance and reduces the risk of conflict between stakeholders. These measures can enhance

the overall stability of the project and avoid negative impacts on the ecosystem, stakeholders, and the local population.

4. In terms of benefit trade-off, it is necessary to make a reasonable distribution of ecological, social, economic, and other benefits based on the results agreed on by stakeholders, and relevant models or frameworks are used to make a balanced distribution of the different benefits brought by the project to ensure that the ecological, social, and economic benefits are all distributed in a balanced manner after the project is implemented. Establish an ecological risk assessment and adaptive management mechanism to identify and respond to possible ecological risks in a timely manner and reduce the likelihood of irreversible damage to the ecosystem.

In summary, by adjusting the implementation plan, we can enhance the stability and sustainability of the "mountain and water project" in the Jiulong River Basin. These measures help ensure that the project has a positive impact on the ecosystem, stakeholders, and local residents while reducing potential risks and conflicts and providing a reference for other landscape projects and watershed restoration projects.

Currently, most ecological restoration projects only evaluate the effectiveness of the project. However, often, problems are found after the project is completed, which requires a higher cost to solve and even misses the opportunity to improve the ecosystem [54]. This study provides an idea for standardizing watershed ecological restoration engineering design and implementation. At the same time, the application of NbS in watershed ecological restoration is not only at the conceptual level. Through this method, it can be judged whether the ecological restoration measures of the watershed are in line with NbS, and specific additional or adjustment suggestions can be given to the scheme according to the assessment results. This process can enhance the overall stability of the project and help the measures bring multiple benefits. Empirical research shows that this method has particular value in the practical application of watershed ecological restoration, but there are still certain limitations. First, in terms of index scoring standards, although this study proposes an idea to calculate the index matching degree score by refining the criteria, the method is still subjective and affected by the scorer's understanding of the implementation plan. Secondly, whether the ecological engineering of a specific watershed can achieve the goals set in the plan is still being determined by other factors, such as the uncertainty of the ecosystem and whether the implementation can meet the plan's standards.

**Author Contributions:** W.L., Y.T. and R.S. co-wrote the paper. W.L. performed the analysis and wrote the paper. W.L., Y.T. and R.S. jointly participated in the evaluation process of the implementation plan. W.L. and R.S. revised the paper and checked grammar and sentence patterns. Y.T. reviewed the manuscript and put forward a lot of practical suggestions in the research process. All authors have read and agreed to the published version of the manuscript.

**Funding:** This study was supported by the National Key Research and Development Project of China (2022YFF1303202).

**Data Availability Statement:** The IUCN Global Standard for Nature-Based Solutions can be accessed through the IUCN Library System: https://portals.iucn.org/library/search/node, accessed on 18 October 2023.

**Conflicts of Interest:** The authors declare no conflict of interest.

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
