# Peer review of "Research on the Application of NbS in Watershed Ecological Restoration: A Case Study of Jiulong River Watershed Shan-Shui Initiative"

_sustainability, doi:10.3390/su152316535_

Round 1
Reviewer 1 Report
Comments and Suggestions for Authors
The article is important when it cope with Nature-based solutions (NbS) watershed restoration. I think it is an excellent contribution to future discussions about sustainability. The idea of evaluate the of watershed restoration projects based on the eight criteria proposed by the IUCN is a nice strategy to call attention to conservation too.

Reviewer 2 Report
Comments and Suggestions for Authors
Overall, this is a very well written manuscript. The information is presented in a logical sequence and there are plenty of references to previous literature that is then used to provide the basis/context for the study. It was a pleasure to review this manuscript and the results.
For the methodology, the "User Guide" is a validated tool that is applicable for use in this study.
Line 188. The following is 187 a specific analysis of the evaluation results(Figure 4). This is correct but the table needs to be brought before the text. I kept trying to figure out "what was the criterion?" It wasn't until I scrolled down that it became clear in the table.
Three recommendations were made in the conclusions section that are accurate.
Reviewer 3 Report
Comments and Suggestions for Authors
Dear Editor-in-Chief of Sustainability
I have some comments about the submitted work
1. Being a proposal to evaluate a standard methodological process of habitat restoration, it would be convenient in the end in the conclusions to evaluate to what extent or what percentage of success is expected after applying the process
2. Being a future project, it needs annual planning tools (AOP) that indicate to what extent the implementation guidelines could be worked on in parallel.
3. From my personal point of view, the process suffers from a diagnostic stage for the territory of the basin, without which there would not be a political-social-ecological framework on which to plan the implementation of the restoration plan, or at least in the document does not refer to the process
4. Finally, there is no reference to the time it would take to implement the restoration plan, which makes it difficult to visualize the project.
Reviewer 4 Report
Comments and Suggestions for Authors
Paper research the application of NbS in watershed ecological restoration on a case study of Jiulong River watershed Shan-Shui initiative. According to the assessment results, the degree of matching based on NbS for the implementation scheme of the Jiulong River Watershed Shan-Shui Initiative is 73 %, which meets the criteria of NbS but needs to be improved in terms of monitoring and assessment, synergistic management, and benefit trade-offs.
General Remarks:
Please comment the novelty of this research in Introduction section.
In Section 5 I would suggest to mention/cite proper references/literature when writing why the research results are important and then to cite where they fit in the current literature...
Specific Remarks:
Change Figures 2 and 4 to Tables or describe them in text.
In Section 4 please put figure of study area research-Jiulong River watershed.
Rows 176-179: two same sentences?
In Section 5 when you mention indicators and criteria, state that they are described in Figure 4.
Row 49: Please put reference.
Row 50, row 121...: Please put reference for the User Guide.
Row 85: Please put reference.
Please commnet some more Figure 2 in text.
In Section 4 you mentioned "sourses"... which are...?
Please commnet some more Figure 3 in text (Index matching degree).
Row 223, 232 and 253: Guidline or Criterion?
Figure 4 presented with colour represention of the results...
Comments on the Quality of English LanguageMinor editing of English language required.
Round 2
Reviewer 4 Report
Comments and Suggestions for Authors
The authors have taken into account the reviewers' comments and suggestions in corrected version of the manuscript, so I suggest that the paper can be accepted in present form.
Comments on the Quality of English LanguageNone